# An integrase toolbox to record gene-expression during plant development

Sarah Guiziou [1], Cassandra J. Maranas[1,2], Jonah C. Chu[1,2] & Jennifer L. Nemhauser [1] ✉

There are many open questions about the mechanisms that coordinate the dynamic, multicellular behaviors required for organogenesis. Synthetic circuits that can record in vivo signaling networks have been critical in elucidating animal development. Here, we report on the transfer of this technology to plants using orthogonal serine integrases to mediate site-specific and irreversible DNA recombination visualized by switching between fluorescent reporters. When combined with promoters expressed during lateral root initiation, integrases amplify reporter signal and permanently mark all descendants. In addition, we present a suite of methods to tune the threshold for integrase switching, including: RNA/protein degradation tags, a nuclear localization signal, and a split-intein system. These tools improve the robustness of integrase-mediated switching with different promoters and the stability of switching behavior over multiple generations. Although each promoter requires tuning for optimal performance, this integrase toolbox can be used to build history-dependent circuits to decode the order of expression during organogenesis in many contexts.

Biologists have long been fascinated by the molecular pathways that support the development of complex multicellular organisms. Plants are particularly intriguing subjects to study, as the development programs that start in their embryos persist throughout their lifespan, strongly influenced by environmental cues. The growing environmental pressures resulting from climate change make this adaptability increasingly important[1]. A better understanding of the mechanisms that underlie plant developmental plasticity will help guide the engineering of traits that can face current and future challenges[2].

To fully understand the molecular trajectory underlying fate transitions that enable de novo organogenesis and regeneration in plants, we need methods that can sense and relay information in a manner that can be dynamically and quantitatively read out by an observer. Current methods enable precise quantification of DNA[3], RNA[4], and proteins[5] allowing the capture of a snapshot of the molecular state of studied organisms. Combining these approaches with single-cell methods has led to the discovery of new plant cell types and a more detailed view of cell-fate transitions[6–12].

A challenge of current single-cell methods is that they require the destruction of samples, and, therefore, do not allow for real-time readouts, reports from the same sample across multiple timepoints, or preserve spatial relationships. With recent advances in high-throughput and high-precision microscopy, fluorescent reporters and sensors have allowed imaging at cellular resolution of transcription level, protein and molecule concentration and localization in a continuous manner in their native context[13]. However, detection is limited to a reduced amount of information at a time due to the limited number of fluorescent tags and to short timescales due to photobleaching and stress to the organisms. Recently, the development of synthetic, DNA-based recording systems has overcome some of the technical challenges of 'omic and microscopy techniques, allowing the sensing and relaying of multiple signals simultaneously during animal development (reviewed in[2]).

Serine integrases, used by bacteriophages to mediate their own integration into the bacterial genome, were critical to the success of one of the most promising synthetic recorders[14]. In a synthetic system,

[1]Department of Biology, University of Washington, Seattle, WA 98195, USA. [2]These authors contributed equally: Cassandra J. Maranas, Jonah C. Chu. ✉e-mail: jn7@uw.edu

serine integrases are used to invert or excise DNA in a site-specific and irreversible manner, referred to here on as an integrase switch. The integrase recognizes two DNA sites of around 40 bp known as attB and attP sites. If the sites are in the same orientation, the DNA region between them is excised and if the sites are in the opposite orientation, the region is inverted. Gene-regulatory parts, such as promoters or terminators, can be placed between integrase sites to mediate a specific gene expression pattern dependent on integrase expressions. Complex genetic circuits have been developed using serine integrases, implementing Boolean logic (in bacteria[15,16], mammalian cells[17], and plant protoplasts[18]), history-dependent logic (in bacteria[19,20]), and cell-lineage tracing (in animals[14]). Serine integrases can also be used to induce the expression of toxic genes at a specific time, to mediate site-specific DNA integration[21]. To date, serine integrases have not been used extensively in plant systems, although they have been shown to work in principle in *Arabidopsis*[22], *Nicotiana benthamiana*[21,23], barley[24], and wheat[25]. One study in *N. benthamiana* used a recombination directionality factor (RDF), which when combined with the integrase, allowed reversing of the integrase reaction[23].

Tyrosine integrases have been more extensively used in Arabidopsis and other plants. Cre recombinase, for example, has been used to perform cell-lineage tracing through DNA excision using a single pair of identical integrase sites[26,27]. Cre has also been used with integrase-site mutants to generate stochasticity in the output[28]. Recently logic circuits have been engineered using DNA excision mediated by a combination of FLP, Cre, and B3RT integrases[29]. Many tyrosine integrases, including FLP and Cre, act on two identical sites, and so, in principle, can catalyze both a forward and reverse recombination reaction[30]. Serine integrases, in contrast, use two distinct sites, leading to a directional and irreversible recombination event. Another limitation on the use of tyrosine integrases in plants has been low efficiency and silencing, which has recently been connected to CHH methylation of tyrosine recombinase sites[31].

Here, we develop a toolbox of well-characterized parts to build synthetic circuits in *Arabidopsis* using PhiC31 and Bxb1 serine integrases. We express integrases from promoters of well-characterized transcription factors essential for lateral root development as a test-case for building synthetic recorders. To optimize the specificity and robustness of our tools when using different promoters, we build and test a variety of methods to tune the threshold for integrase switching-

tools that could be used to tune the activity of any protein of interest. Finally, we characterize two methods that allow for further fine-tuning of the timing and level of integrase activity: split-intein integrase and estradiol-inducible integrase. Collectively, these modular parts make it possible to record gene expression at specific times and spaces during plant development, as well as contribute to an accelerated design-test-build-learn cycle for other plant synthetic biology devices.

## Results
### Orthogonal and efficient DNA switches in *Arabidopsis*
Our first goal was to test the efficiency of three serine integrases (PhiC31, Bxb1, and Tp901) in *Arabidopsis* transgenic lines. To do this, we needed two constructs: the target and the integrase. These two constructs cannot be on the same plasmid, as even low levels of integrase produced in bacteria during cloning will enable target site inversion. For our target construct, we used a constitutive promoter (pUBQ10[32];) flanked by integrase sites positioned between two reporter genes: the mScarlet and mTurquoise2 (mTurq) fluorescent proteins (Fig. 1a). To more closely match the expression level of most developmentally-relevant genes, we opted to use the promoter of PROTEIN PHOSPHATASE 2 A SUBUNIT 3 (pPP2AA3), a gene which has been widely used as a qPCR control due to its constitutive nature and medium expression level[33]. If expressed and functional, the integrase should mediate the inversion of the promoter resulting in the switch of expression from mTurq to mScarlet. Targets containing either PhiC31 or Bxb1 integrase sites strongly expressed mTurq and not mScarlet in roots and leaves (Fig. 1b). When either the PhiC31 or Bxb1 integrase was constitutively expressed alongside the target with its cognate integrase sites, we observed exactly the opposite reporter expression (strong mScarlet and no mTurq) in all tissues. Moreover, we confirmed that PhiC31 and Bxb1 integrases are orthogonal to one other, as Bxb1 integrase does not mediate an integrase switch in the PhiC31 target line nor vice versa (Fig. 1c). The Tp901 integrase, known to be less efficient than Bxb1 and PhiC31[34], did not cause any switching in targets carrying its target sequences, even with strong promoters (p35S and pUBQ10), and codon optimization (Supplementary Fig. 1). We also tested a switch using YFP and Luciferase reporters that had been used previously in *N. benthamiana*[23], and confirmed that the pPP2AA3 promoter allows constitutive integrase switch with this target as well (Supplementary Fig. 2).

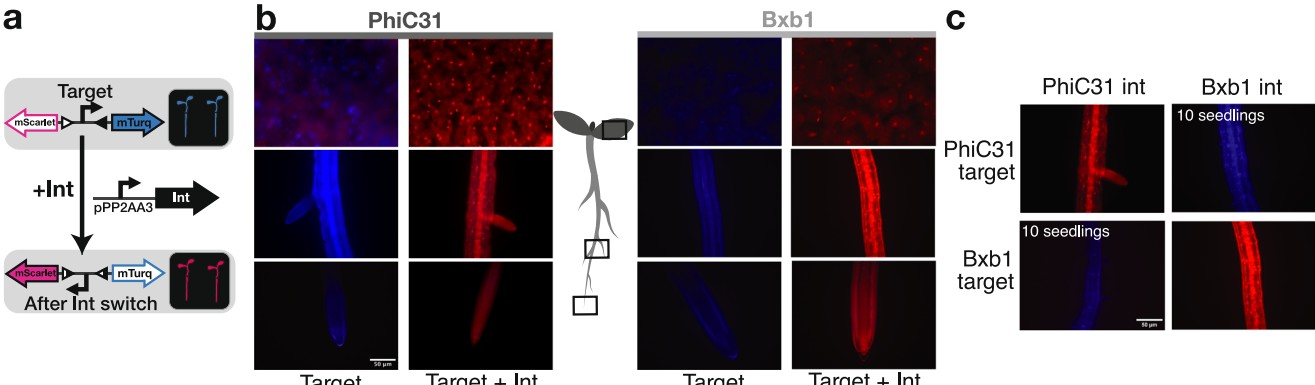

**Fig. 1 | Integrase mediates orthogonal DNA-switch in *Arabidopsis*. a** Design of the integrase target. The target is composed of two integrase sites (triangles) surrounding a constitutive promoter (pUBQ10) and two fluorescent reporters (mTurquoise2 and mScarlet). In absence of integrase, mTurquoise2 is expressed. In presence of integrase, the integrase mediates inversion of the DNA between the integrase sites, inverting the promoter, and leading to mScarlet expression. The expression of the integrase is mediated by the constitutive promoter pPP2AA3. **b** Constitutive integrase switch characterization. On the left side, *Arabidopsis* seedlings with PhiC31 target alone and PhiC31 target with pPP2AA3:PhiC31

construct. On the right side, Bxb1 target alone, and Bxb1 target with pPP2AA3:Bxb1 construct. Microscopy images are an overlay of mTurq (in blue) and mScarlet (in red) fluorescence, from top to bottom are representative images of the leaf, a lateral root, and the root tip ($n = 20$ seedlings screened). The scale bar in the left bottom image applies to all images. **c** Orthogonality test of the integrase switch. Each integrase target line was transformed with both integrases. Microscopy images are overlays of mTurq (in blue) and mScarlet (in red) fluorescence, and are representative images ($n = 10$ seedlings screened). The scale bar in the left bottom image equals 50 μM and applies to all images.

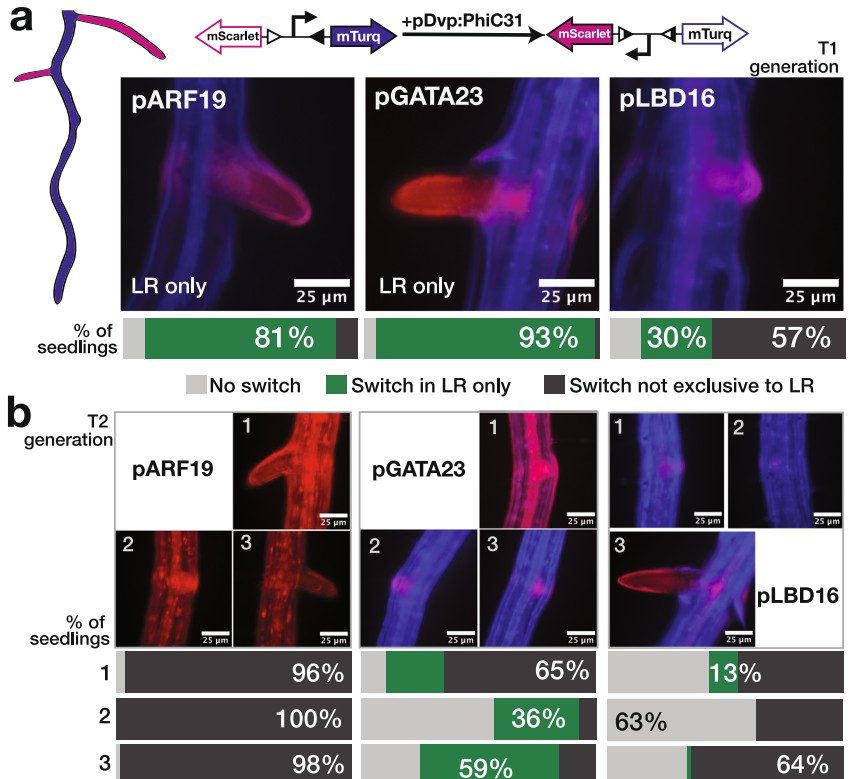

**Fig. 2 | PhiC31 integrase switch under the control of developmental promoters.** **a** Developmental (Dvp) promoters drove the expression of PhiC31 integrase with a target that switches from mTurquoise2 to mScarlet when the integrase is active. Target lines were transformed with the integrase constructs, and at least 20 T1 seedlings per integrase constructs were characterized. Representative images of emerged lateral roots are shown for each promoter-integrase construct, as well as a bar representing the percentage of seedlings in each phenotypic category: no switch (light gray), switch in LR only (green), or switch not exclusive to LR (dark gray). **b** Characterization of T2 seedlings. For each construct, we selected 3 T1 lines with an LR-only switch phenotype, and characterized 20 T2 seedlings per T1 line. For each T1 line, a representative T2 seedling is shown above bar graphs displaying the percentage of seedlings in each phenotypic category. The percentage of the phenotype represented by the T2 image is displayed numerically on the relevant portion of the graph. Source data are provided as a Source Data file.

To record developmental events, switching of the integrase targets must be consistently restricted to a narrow range of time and space. For the next round of design and testing, we focused on the PhiC31 integrase and lateral root development, a well-characterized example of de novo organogenesis[35]. Lateral root development is a good model for applying synthetic tools to study gene expression because it is a well-studied pathway with defined transcriptional control points[36]. The density and placement of lateral roots are also features of plant architecture that are linked to climate resilience and therefore a strong candidate for synthetic engineering[37]. Lateral roots initiate from a small population of founder cells at the xylem pole of the pericycle layer[38], and follow a fairly stereotyped pattern through morphogenesis[39].

As test drivers for integrase expression, we selected the promoters of several well-studied transcription factors expressed in the early stages of lateral root initiation: AUXIN RESPONSE FACTOR 7 (ARF7)[40], AUXIN RESPONSE FACTOR 19 (ARF19)[40], LATERAL ORGAN BOUNDARIES DOMAIN 16 (LBD16)[41], and GATA TRANSCRIPTION FACTOR 23 (GATA23)[42]. Because the integrase switch is heritable, we would expect that if the integrase is expressed in lateral root founder cells and works efficiently, all cells in the new root should be in the switched state as well. Simply put, all of the cells in the main root should express mTurq, while all of the cells of the lateral roots should express mScarlet. We characterized approximately 20 independent transformants (T1s) per integrase construct, and categorized each seedling by the following categories: (1) No-switch: expression of mTurq only; (2) LR-only: expression of mScarlet in lateral root only; or (3) Non-exclusive: any expression of mScarlet in the main root. For pARF19 and pGATA23, a majority of the T1s was switched in the lateral

root only (81% and 92% of the seedlings, respectively) (Fig. 2a), showing that the integrase switch can record the transcription of a development-related gene. Additionally, this data prove that the integrase system can faithfully trace cell lineage, as all cells, even those in fully emerged lateral roots, continued to express mScarlet only.

Other promoters did not fulfill the specifications. For pLBD16, we observed only 30% LR-only seedlings, while 57% of T1s showed non-exclusive expression of mScarlet (Fig. 2a). Most of the seedlings in the non-exclusive category (70%) did not display switching in the entire seedling, but instead had mScarlet expression in a few cells in the vasculature in addition to the lateral root (Supplementary Fig. 3). This "weak" non-exclusive switching pattern (Supplementary Fig. 3) corresponds to the known expression pattern of LBD16[43–45]. For ARF7, 79% of seedlings were switched in all tissues of the root (Supplementary Fig. 4). Roughly half of the non-exclusive seedlings showed a full switch and half showed some expression of mTurq (in addition to mScarlet) in the entire root. This result is consistent with ARF7 being expressed in other tissues and other times of development[46].

Our next question was whether the integrase system would remain robust over subsequent generations, or whether some low level of leakiness would lead to plants where every cell was in the switched state. For these experiments, we selected three T1 lines where PhiC31 was driven by pARF19, pGATA23 or pLBD16, and which were characterized as having LR-only switching. From each line, we characterized 20 progeny (T2 seedlings). In all cases, we observed a decrease in LR-only seedlings in the T2 generation (Fig. 2b). For pGATA23 T2s, we observed LR-only switches, but at a lower proportion than in T1s, and obtained seedlings displaying no-switch and non-exclusive switches. This pattern is not surprising, as in the T1

generation, plants are hemizygous for the integrase transgene insertion events, meaning that some T2s may end up with no integrase and others may have different numbers of insertions leading to a range of expression levels. For *ARF19*, the majority of T2 seedlings are fully switched (96%, 100%, 98%), also consistent with an increased dosage of integrase in many of the T2s. The lack of no-switch category T2s for this construct suggests that the integrase expression may be happening during gamete development and then transmitted to all of the cells in the T2 generation. For pLBD16 T2s, most of the seedlings are either no-switch or non-exclusive, with 66% or more of the non-exclusive seedlings having a weak switch in the main root similar to the T1 generation (Supplementary Fig. 3). To obtain a robust, cell-type specific switch, the expression level of the gene of interest in those cells should be significantly greater than that in other cell types. With *LBD16*, it seems that the expression level in LR cells is similar enough to that in the phloem pole pericycle (BAR Webservices[43],) to make LR-only switching rare.

### A suite of tools to optimize switch sensitivity by tuning integrase activity

The integrase switch is a binary output, while gene expression is analog and conventionally defined relative to a standard "background" or "basal" level. For example, low-level gene expression is often "rounded down" to be defined as off when it falls below an arbitrary threshold and is considered specific to a developmental event when enriched above a similarly arbitrary threshold. To be able to record events marked by different promoters, each with their own relative levels of "off" and "on", we needed to be able to tune the sensitivity of the integrase switch (e.g., at what level of promoter expression the integrase switch is activated; Fig. 3a). While there is a rich literature of characterizing modular modifications for tuning protein activity in other systems, there are relatively few such parts available for plant synthetic biologists. We decided to characterize modifications that were predicted to work at the transcriptional, translational, and post-translational levels (Fig. 3b).

We tested our tuning methods by expressing the integrase construct constitutively and observing the resulting level of switching in the roots of T1 seedlings. While in theory, the constitutively expressed integrase should cause every cell to behave in the same manner, stochasticity in transcription, translation, and integrase activity results in cell-to-cell variation in the precise timing when switching occurs. This variation makes it possible to use the level of switching at a given time point as a performance metric that serves as a proxy for integrase activity. To capture the range of variation observed, each seedling was assigned to one of five classes, capturing the relative level of mTurq to mScarlet observed (Fig. 3c, Supplementary Fig. 5). The classes ranged from no switching (mTurq only) to full switching (mScarlet only).

To mimic integrase expression under developmental promoters of different strengths and to capture the impact of transcriptional control modifications, we used three constitutive promoters of increasing strengths: pPP2AA3, pUBQ10, and p35S (Fig. 3d). We observed subtle differences in the switching behavior among the various promoters with p35S-driven integrase lines showing the highest percentage of seedlings in the full-switch class (Fig. 3d). As a further test of transcriptional control and in recognition of recent work documenting the striking impact of terminator sequences on gene expression[47–51], we switched the UBQ1 terminator for one of our promoters, pPP2AA3, to the 35 S terminator. We found that the constructs with t35S showed a decreased switch sensitivity compared to those with tUBQ1 ($p < 0.001$) (Fig. 3e). This result further highlights the importance of promoter-terminator interactions, which could involve loop formation or the preferential localization of transcription factors to different terminator regions[52,53].

For post-transcriptional modifications, we studied the impacts of an SV40 T antigen-derived nuclear localization signal (NLS)[54] predicted to increase integrase activity[55] and an RNA destabilization tag (DST) from SMALL AUXIN UP-REGULATED RNA (SAUR) genes[56] predicted to decrease activity (Fig. 3b). For pPP2AA3, the addition of the NLS appeared to increase the proportion of fully switched seedlings when compared to the construct with the integrase alone, although the difference was not statistically significant ($p = 0.15$) (Fig. 3d). For the stronger promoters pUBQ10 and p35S, the addition of the NLS did not significantly affect the switching threshold, which was likely already at a maximum level. The addition of the DST significantly decreased the switch sensitivity for all three promoters (pPP2AA3: $p < 0.01$, pUBQ10, p35S: $p < 0.001$) (Fig. 3d). In all cases, the addition of the DST increased the proportion of seedlings in the weaker switch categories and, in the case of pUBQ10 and p35S, reduced the proportion of fully switched seedlings.

As a final option for post-translational tuning, we tested two ubiquitin (Ub)-based protein destabilization tags (Fig. 3e). These tags work by exposing an N-terminal residue which triggers the degradation of the protein by ubiquitin ligases[57]. Previously characterized in *Saccharomyces cerevisiae* (yeast), the N-end rule states that the identity of the N terminal residue determines the half-life of the protein, thus different Ub degrons confer varying levels of instability[58]. We chose a Ub-Arginine (UbR) and a Ub-Glutamine (UbQ) degron to test as in yeast they had a strong or modest impact on protein turnover, respectively[59]. While the N-degron pathway has been characterized in plants[60], it has not been used in synthetic circuits *in planta* to tune protein levels. Consistent with the yeast results, we found that both degrons significantly increased the threshold for the integrase switch when compared to the integrase alone ($p < 0.001$ for both comparisons), with UbR acting more strongly than UbQ.

As transient expression in *N. benthamiana* is a favorite testbed for plant synthetic biology applications, we wanted to know whether this toolbox of tuning strategies would be useful in that context as well. In addition to the pPP2AA3, p35S, and pUBQ10 promoters, we also tested the collection of tuning options with pARF19, classified here as a weak constitutive promoter. In addition to its well-known role in initiating new lateral roots, ARF19 is also important for leaf expansion in *Arabidopsis*, so we hoped that it might be expressed in *N. benthamiana* leaves as well[61]. The NLS had a similar effect as in *Arabidopsis*, increasing the switch sensitivity for the integrase expressed under pPP2AA3 and pARF19 but not pUBQ10 or p35S (Supplementary Fig. 6). Unlike in *Arabidopsis*, the DST did not have a significant effect on switching in *N. benthamiana* (Supplementary Fig. 6). The 35S terminator with the NLS significantly increased switching and without the NLS tag increased switching with approaching statistical significance ($p = 0.17$) as compared to the UBQ1 terminator in *N. benthamiana* (Supplementary Fig. 6). This is in contrast to our findings in *Arabidopsis*. In addition, the effect of the Ub degrons was quite different from what was observed in *Arabidopsis* (Supplementary Fig. 6). UbR, which drastically reduced switching in *Arabidopsis*, did not significantly affect the switching in *N. benthamiana*. Even more surprisingly, UbQ which conferred a more modest, but significant reduction in switch sensitivity in *Arabidopsis*, increased switching in *N. benthamiana*. These differences have practical implications for optimizing synthetic devices, but also point to potentially fundamental differences in N-end rule dynamics and control between the two plants.

### Engineering robust integrase switches with developmental promoters

We next wanted to test the impact of the tuning modifications for developmental promoters, and focused on the impact of the NLS and DST in combination with pGATA23 and pARF19 constructs. As expected, the NLS increased the proportion of seedlings showing non-exclusive switching from less than 10% to above 86% (Fig. 4a). Conversely, the addition of the DST led to the absence of non-exclusive

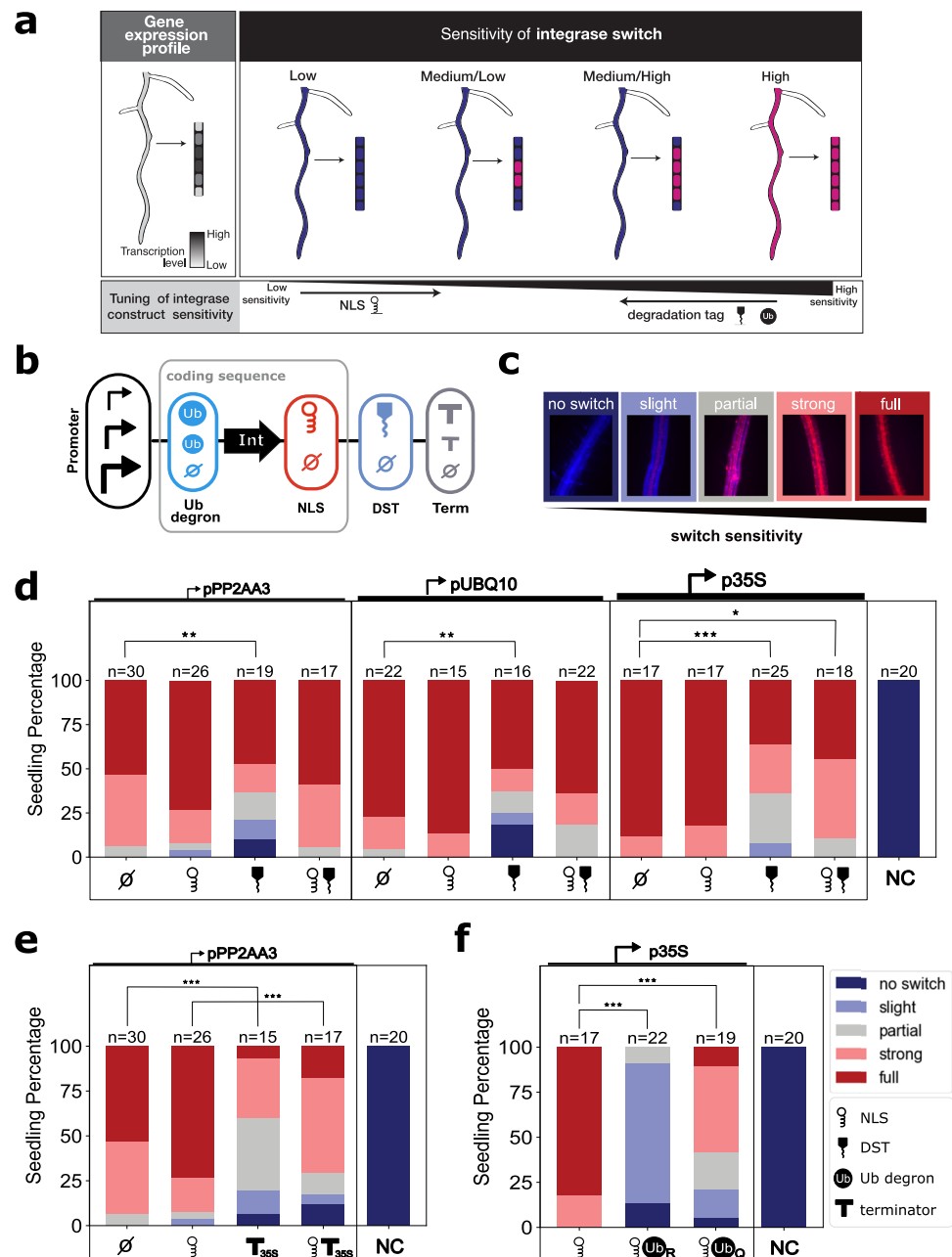

**Fig. 3 | Methods for tuning integrase switch sensitivity. a** For the gene expression profile of a given gene, a low-sensitivity integrase switch will result in little or no switching in any cells while a high-sensitivity switch will result in complete switching even in cells with relatively low expression of the gene. Different sensitivities of the integrase switch can lead to switches occurring at different levels of transcription. This sensitivity must be tuned to achieve the desired specificity for a given gene expression profile. **b** The integrase tuning constructs consist of a constitutive promoter controlling the integrase expression with tuning add-ons including a nuclear localization signal (NLS), RNA destabilization tag (DST), ubiquitin (Ub) degrons, and varied terminator. **c** The level of sensitivity is sorted into one of five categories, evaluated based on the level of mScarlet compared to

mTurquoise fluorescence (Supplementary Fig. 5). **d**, **e** To evaluate statistical significance, each switching category was assigned a number from 1 through 5 (1 = no switch, 5 = full switch). Significance was determined using a two-sided analysis of variance (ANOVA) with a post-hoc Tukey's Honest Significance Difference test (*$p < 0.05$, **$p < 0.01$, ***$p < 0.001$). The negative control (NC) is the target line without any transformed integrase construct. **d** Tuning results using 3 constitutive promoters (pPP2AA3, pUBQ10, p35S) with NLS, DST, and both. From left to right, the *p* values are as follows: 0.0099, 0.0057, 0, 0.0235. **e** Tuning results from varying the terminator from tUBQ1 to t35S. *p* values from left to right are 0.0001 and 0.0017. **f** Tuning results from and Ub degrons with all *p* values equal to 0. Source data are provided as a Source Data file.

switching, and a slight increase in seedlings with no observed switch (from 5% no-switch in pGATA23 alone to 9% with DST; from 9% no-switch in pARF19 alone to 10% with DST) (Fig. 4a). Similarly, for pLBD16, the addition of the NLS leads to non-exclusive switching in 100% of seedlings, with 97% of the seedlings fully switched to mScarlet expression in the entire seedling (as opposed to the small number of non-LR cells showing switching in the transgenics expressing

unmodified integrase from pLBD16) (Fig. 4a and Supplementary Fig. 3). When pLBD16 was used to drive expression of an integrase modified with the DST, no seedlings were categorized as LR-only (Fig. 4a), but there were a higher proportion of seedlings with a switch only in the LR and in a few cells in the main root (76% with DST, 47% without; Supplementary Fig. 3). This trend is consistent with the DST allowing recording of which cells express the highest level of *LBD16*.

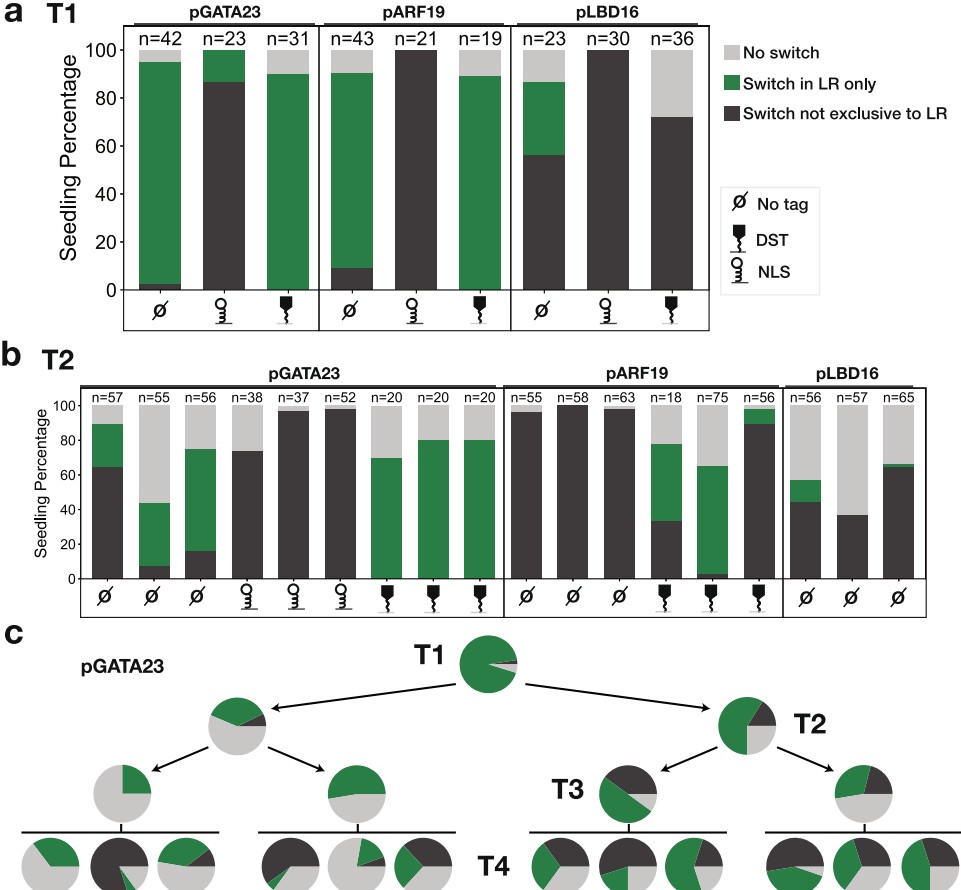

**Fig. 4 | Developmental promoters with tuning tags are stable over multiple generations. a** Phenotyping of T1 seedlings with constructs in PhiC31 target line, PhiC31 integrase is expressed from the indicated developmental promoters in combination with various tuning tags (legend on the right). The percentage of seedlings in each of the defined phenotypic categories is shown. **b** Phenotyping of T2 seedlings from a subset of the T1 lines represented in (**a**). T2 from three T1 lines per construct were characterized, all the T1 lines selected for T2 were switching only in the LR. **c** Percentage of pGATA23:PhiC31 (no tag) seedlings in each phenotypic category over four generations. The pie charts for T1 and T2 are derived from the same data as displayed in (**a**) and (**b**). From each generation, three seedlings categorized as LR-only were selected for propagation. Source data are provided as a Source Data file.

As for the constructs without tags, we tracked the stability of integrase switching over multiple generations. As seen previously, the ratio between switching categories changed somewhat between T1s and T2s. For pGATA23, the addition of the DST increased the stability of the phenotypic ratio of the T2 generation (Fig. 4b). We did not observe any T2 seedlings with non-exclusive switches while using the DST. The presumed increase in integrase efficiency with the NLS led to a complete absence of T2 seedlings with an LR-only switch, consistent with the T1 pattern. For pARF19, the integrase switch was no longer LR-specific in T2s (Fig. 2b). The addition of the DST increased the stability of the switch in the T2 generation, leading to an LR-only switch of up to 47% of seedlings in one T1 family (Fig. 4b). We still observed a high variability between families, with some showing mostly non-exclusive switching (Fig. 4b and Supplementary Fig. 7). For pLBD16, the T2s had a higher proportion of non-exclusive seedlings, although the addition of the DST narrowed the extent of mScarlet expression outside the LR (Fig. 4b and Supplementary Fig. 3). We also investigated the extent to which no-switch T2s represented individuals that had lost the integrase (as T2s were not selected on an antibiotic before characterization). After performing a post-characterization selection, we found that the proportion of no-switch seedlings was highly reduced (Supplementary Fig. 8), meaning that we are likely underreporting the stability of the lines in T2s.

In summary, tuning allowed us to obtain at least one T1 line for each promoter that accurately recorded the expression of the corresponding native gene. As mentioned previously, approximately one quarter of T2 seedlings from each T1 line lost the integrase construct

via segregation and therefore was not switched. These unswitched seedlings can be easily removed from the further analysis without requiring the use of antibiotic selection, as we did not see switching in the absence of integrase in any case; however, this background could be an issue for some applications.

To analyze later generations, we followed a T1 carrying pGATA23:PhiC31 (no tag) over four generations, propagating two LR-only seedlings at each generation. In the fourth generation, we obtained in median 32.5% (ranging from 5 to 60%) of seedlings with an LR-only phenotype (Fig. 4c). While there was clear line-to-line variability and some loss of phenotypic robustness, we could find many lines where the integrase-based recorder was still working well even in the T4 generation. The addition of the DST appeared to further stabilize the recorder function, as 65 to 100% of T3 seedlings were LR-only and there were no seedlings with non-exclusive switching phenotypes (Supplementary Fig. 7c).

In addition to wanting performance stability across generations, we also wanted to make sure that an integrase-based recorder would faithfully record the spatiotemporal pattern of developmental gene expression, as there is an inherent lag between the induction of a promoter of interest and the time when the switched target reporter is detectable. To test whether this time gap was relevant to the timescales of lateral root development, we compared the expression pattern of our genes of interest using a traditional transcriptional reporter with the expression pattern of the integrase switch system driven by the same promoter. While our initial characterization shown in Fig. 2

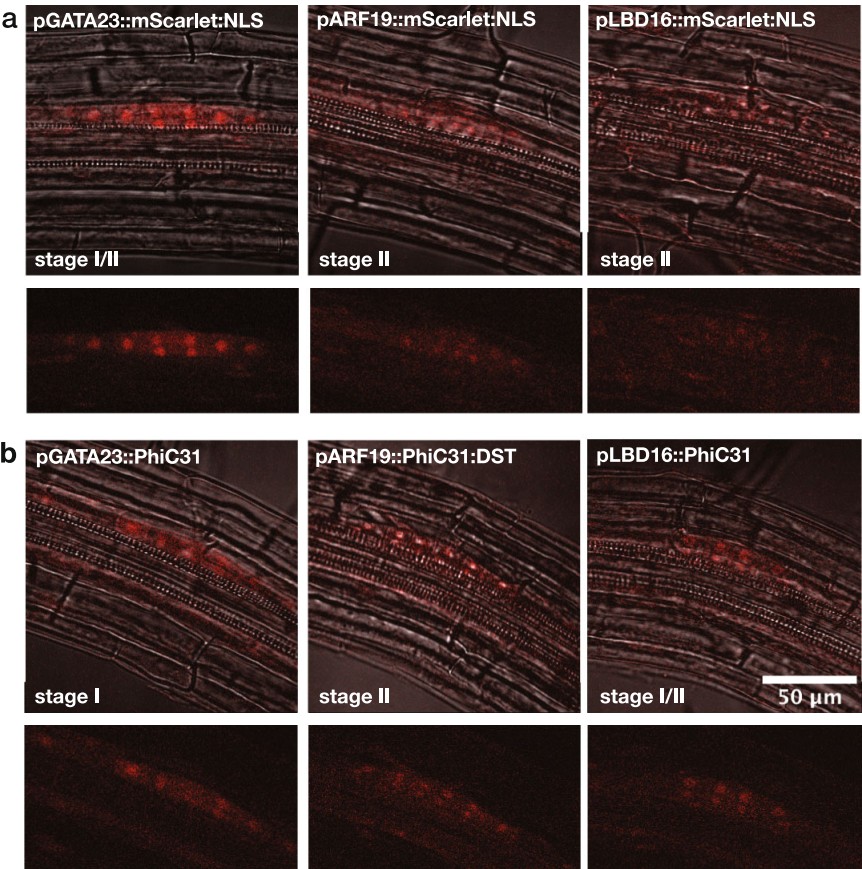

**Fig. 5 | Confocal imaging of transcriptional reporters and integrase-based recorder in early-stage lateral roots. a, b** Overlays of brightfield and red fluorescence channels from a single frame are shown above an image of the red fluorescence channel alone. For each image, the developmental stage of the lateral root primordium is indicated. **a** Transcriptional reporter lines, composed of the promoter of interest driving expression of mScarlet fused to an NLS. **b** PhiC31 integrase-based recorder lines (pDvp:PhiC31 in PhiC31 target line switching from mTurq to mScarlet) with any modifications indicated in each panel. 3 seedlings per line were imaged. Source data are provided as a Source Data file.

revealed the overall pattern of integrase-based recorder activity, for these comparisons we focused our attention at the earliest stage of lateral root development. Onset of expression for transcriptional reporters and integrase constructs appeared essentially identical (Fig. 5), indicating that the integrase system records the spatio-temporal pattern of gene expression with no significant delay. Beyond allowing for heritable gene expression in all daughter cells, an additional benefit of the integrase system was an amplification of the developmental promoter signals. This was most obvious with the weakest promoter, pLBD16. By the same logic, the integrase system could be of great use for any application requiring normalization of output levels from multiple promoters or across multiple input signals.

**Increasing the potential applications of the integrase-based recorder**
Another challenge with the integrase-based recorder is that many cellular events of potential interest may not have well-characterized promoters associated with them, or may rely on promoters that are activated in multiple cell types or conditions. For example, any promoter active in the embryo could trigger the switch of the integrase target, making any subsequent recording impossible. To overcome this limitation, we built additional tools that allow induction of the integrase at a user-determined time in development.

The first of these tools is the split intein integrase system which has already been characterized in vitro[62]. Inteins are sequences that trigger autocatalytic splicing, making it possible to reconstitute proteins from fragments expressed from two separate constructs[63], a technique that has been used previously in plants[64]. In the split intein integrase system, we applied here, the PhiC31 integrase is split into two extein domains: the N-terminal sequence fused to the intein N-term: Npu DnaEN and the PhiC31 C-terminal sequence fused to the intein C-term: Ssp DnaEC (Fig. 6a). Expression of the two components in a single cell triggers post-transcriptional trans-splicing, generating a fully functional PhiC31 integrase. We tested the split-intein integrase system in *Arabidopsis* with the PhiC31 integrase using strong constitutive promoters for the expression of the two components: pUBQ10 for the N-term, and p35S for the C-term, and found that it worked well (Fig. 6b). We compared the level of integrase switch from this construct to the full integrase under the control of each of the constituent promoters, and found that the split-intein system led to a decrease in switch efficiency. While 90% of split-intein T1 seedlings showed some level of switching, no full switch was observed. In contrast, when the full integrase was expressed under the control of either pUBQ10 or p35S, 75% or more of the seedlings were fully switched. This is consistent with reports that the trans-splicing approach delays the integrase switch in *E. coli*[62]. We additionally tested the split-intein integrase system with pARF19 driving both integrase components (Supplementary Fig. 9a). We observed LR specific switches in T1 seedlings (Supplementary Fig. 9b). Even with its lower switching efficiency, the split-intein integrase system allows the recording of developmental gene expression. Using pARF19:Int without tags led to most T2 seedlings having non-specific switching outside of the lateral root. By combining pARF19 with the split-intein system, most of the seedlings now showed the desired LR-specific switching phenotype

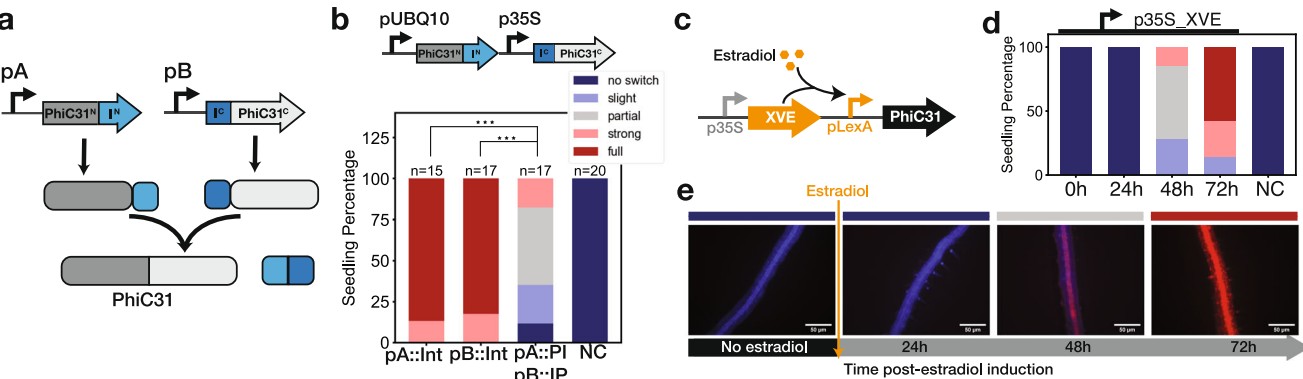

**Fig. 6 | Split-intein and inducible promoters as additional tools to tune and induce the integrase switch. a** Schematic of the split-intein integrase system. The system is composed of two constructs: (i) promoter A driving the N-terminal half of the integrase (PhiC31[N], dark gray) fused to the N-terminal portion of the intein protein (I[N]: Npu DnaE[N], light blue) and (ii) promoter B driving the C-terminal portion of the intein (I[C]: Ssp DnaE[C], dark blue) fused to C-terminal half of the integrase (PhiC31[C], light gray). When the two constructs are expressed, the inteins auto-catalyze trans-splicing, covalently joining the two parts of PhiC31. **b** The split-intein system reduces the efficiency of the integrase switch. Following the nomenclature of (**a**), pUBQ10 = promoter A and p35 = promoter B. The integrase switch efficiency of the split-intein system is compared with the full integrase expressed with promoter A alone (pUBQ10: PhiC31) and with promoter B alone (p35S:PhiC31). NC corresponds to the target line without integrase. T1 seedling phenotypes are determined with fluorescent microscopy images and categorized from no switch to full switch. The data were tested for significance using an two-sided ANOVA and post-hoc Tukey's HSD test (*$p < 0.05$, **$p < 0.01$, ***$p < 0.001$), with all $p$ values equal to 0. **c** The estradiol inducible integrase construct is composed of p35S:XVE (transcriptional activator composed of a DNA-binding domain of LexA, the transcription activation domain of VP16, and the regulatory region of the human estrogen receptor[66];) and pLexA-minimal 35S driving expression of PhiC31. **d** Characterization of the estradiol inducible integrase construct shows induction as early as 48 h after treatment. 7 T1 Seedlings with the estradiol integrase construct in the PhiC31 target line were characterized just before estradiol treatment and every 24 h following. The bar graph represents the percentage of seedlings with a given level of switching (classes are color coded as in (**b**), $n = 14$ seedlings. **e** Representative images of a seedling at the specified time point relative to estradiol induction. Source data are provided as a Source Data file.

(Supplementary Fig. 9c). In this way, the split-intein system can be used as a tuning mechanism, as well as allowing the combination of different promoters.

The split-intein integrase system could be used to induce the integrase recording system at a specific stage of seedling development, thereby avoiding recording at earlier stages. This would be done by placing one component under the control of a developmental promoter and the other under the control of an inducible promoter. We could then activate the integrase system through the inducible promoter at the beginning of an experiment to record the expression of genes only after a specific time point, reducing issues with genes expressed in embryonic tissues. As a proof-of-principle for this design, we used the heterologous estradiol-inducible system[65,66] to drive integrase expression. This same system has been used to induce the expression of Cre recombinase in the context of cell-lineage tracing[26–28]. Before estradiol induction, we did not observe any switch, confirming that the estradiol system had an undetectable level of background activity (Fig. 6c). After induction, we analyzed seedlings every 24 h and observed the earliest signs of switching at 48 h with more than 50% of seedlings fully switched by 72 h. This timing fits well with reports that estradiol induction of a reporter peaks at 24 h[66], and would suggest that it takes approximately 24 h after promoter activation for the integrase to become active, mediate the switch and then allow expression and maturation of mScarlet. We also characterized our inducible integrase construct in T2s and confirmed that the baseline of expression without inducer is low enough to prevent any integrase switching in subsequent generations (Supplementary Fig. 10).

## Discussion
Integrase-based recorders of gene expression have a number of advantages over current methods of tracking transcription in individual cells. Among the most prominent of these is that early events can be read-out much later in development, and, in the designs presented here, there is no need to disrupt the spatial relationship between cells. Even more, the use of serine integrases over tyrosine integrase brings new design possibilities to build history-dependent trackers and

complex integrase-based devices. We have added a suite of characterized parts to help synthetic biologists build serine integrase devices in *Arabidopsis*. In addition to the PhiC31 and Bxb1 integrases and cognate targets, we built and tested tuning modifiers like RNA and protein destabilization tags and a split-intein control module. This entire suite of standardized tools can be directly implemented in any system where fine control of protein levels is needed to optimize performance. We also provided proof-of-principle that integrase-based recorders can be used to capture the history of gene expression at specific times and spaces during plant development. While we observed line to line variability in the robustness of our integrase switch, we successfully obtained at least one T1 integrase reporter line per promoter which faithfully records the spatiotemporal pattern of the gene expression. Importantly, we also found that switches functioned robustly over multiple generations. In all these experiments, we set a high standard for transparency around performance variability in plant synthetic devices, including showing quantitative characterization of many seedlings from different independent insertion events over several generations. Additionally, we observed differences in the effect of tuning parts between *Arabidopsis* stable lines and *N. benthamiana* transient assays, adding another note of caution in developing synthetic devices for use in multiple plants.

The integrase-based recording system characterized here can be readily adapted to the tracing activity of other promoters, including those expressed in other tissues and developmental processes. Because the integrase acts as a signal amplifier, the integrase system could be of interest in the following expression of any genes that are difficult or impossible to observe with traditional reporters. Additionally, the integrase system could be used to record the expression of genes in situations where live imaging is not available. For example, while many labs have at least some access to fluorescent microscopy, most do not have sophisticated live-imaging setups. There are also conditions, such as roots growing in natural soil conditions, where it would be highly advantageous to read out early expression events much later in development. Moreover, in situations where imaging is not compatible with other protocols (e.g., some fixation techniques),

it is also possible to detect the state of the integrase targets used here by sequencing.

Additional synthetic devices should now be accessible working from the toolbox described here. For example, one challenge in producing a developmental recorder is that many promoters of interest are expressed at multiple points in development. One solution would be to combine our inducible integrase and split-intein integrase system, where one part of the integrase is under the control of the externally inducible promoter, and the other is expressed from the developmental promoter of interest. Another use of the split-intein integrase would be to use it as an AND gate by placing the two split-intein components under the control of two promoters from genes of interest. This will allow the recording of when and in which cells two different genes are simultaneously expressed. By using both PhiC31 and Bxb1 integrases, a history-dependent tracker could be constructed with the capacity to record on a single cell level if, and in what order, two genes are expressed. A similar design has been shown previously to work in bacteria[19].

In addition to contributing to our understanding of existing organisms, integrase-based devices can also enable the engineering of novel forms or functions by driving the expression of genes other than reporters. For example, integrase switches could be used to induce the expression of a toxic gene under certain conditions. Cre recombinase, a tyrosine integrase, has already been used to generate homozygous fertilization-defective mutants in plants[67], and to activate a large-tumor antigen in mice[68,69]. A particularly exciting application to imagine is to replace reporter genes in integrase targets with transcription factors able to initiate entire response cascades. Root development could be re-coded by implementing history-dependent synthetic signaling circuits that used integrases to activate developmental regulators, potentially helping plants survive drought or flooding.

## Methods
### Construction of plasmids
Our cloning strategy was based on Golden Gate assembly using appropriate spacers (Supplementary Fig. 11)[70] and BsaI-HFv2 (NEB) as the restriction enzyme. Candidate promoter sequences (ARF7: AT5G20730, ARF19: AT1G19220, LBD16: AT2G42430, GATA23: AT5G26930) were amplified from Col-0 genomic DNA to add specific Golden Gate spacers. After gel purification, each level0 promoter sequence was cloned using a Zero Blunt PCR Cloning Kit (Thermo-Fisher Scientific). The PhiC31 integrase sequence was a gift from the Orzeaz lab. Bxb1 and Tp901 sequences were a gift from the Bonnet lab. Integrases were amplified using primers with golden gate compatible spacers to generate level 0 integrase parts (primer list available in Supplementary Data 1). Constitutive plant promoters and terminators were purchased from Addgene as part of the MoClo Toolbox for Plants[70]. Some level0 parts were ordered from Twist Bioscience: a mutated version of the pPP2AA3 promoter without BsaI sites, the DST, the Ub-tags, and the mTurq-tUBQ10 level0 construct for target construction. The mScarlet-tRBCs level0 construct was amplified from a transcriptional reporter[10]. Other level0 fragments were ordered from IDT as Gblocks: the codon-optimized Tp901 integrase sequence, the two split-intein PhC31 constructs, and the integrase target sequences without promoters. For the integrase target level0 sequences, the pUBQ10 promoter was added by Golden Gates using BbsI sites.

Construction of constitutive and lateral root-specific level 1 integrase constructs was performed via Golden Gate reaction in the modified pGreenII-Hygr vector containing compatible Golden Gate sites[71]. The construction of integrase targets was performed with the same methods in a modified pGreenII-Kan vector. Construction of level 2 integrase constructs, such as the split-intein system construct, was performed by amplifying completed level 1 integrase constructs using primers with golden gate compatible spacers, then performing Golden Gate reactions in the modified pGreenII-Hygr vector containing

compatible Golden Gate sites. Construction of promoter reporters was performed by assembling through Golden gate reaction the mScarlet with NLS, tRBCs terminator, and promoter in the modified pGreenII-Hygr vector[10]. Details on constructs and primers can be found in Supplementary Data 1 and 2.

Enzymes for Golden Gate assembly were purchased from New England Biolabs (NEB, Ipswich, MA, USA). PCR was performed using 2X Q5 PCR master mix (NEB) and GoTaq master mix for colony PCR (Promega, Madison, WI, USA). Primers were purchased from IDT (Louvain, Belgium), and DNA fragments from Twist Bioscience or IDT. Plasmid extraction and DNA purification were performed using Monarch kits (NEB). Sequences were verified with Sanger sequencing by Azenta Life Sciences (Seattle, USA). Chemically-competent cultures of the *E. coli* strain DH5alphaZ1 (laciq, PN25-tetR, SpR, deoR, supE44, Delta(lacZYA-argFV169), Phi80 lacZDeltaM15, hsdR17(rK −, mK +), recA1, endA1, gyrA96, thi-1, relA1) were transformed with plasmid constructs containing kanamycin resistance. Transformed *E. coli* was grown in LB media (LB broth, Miller) with kanamycin (Millipore Sigma, 50 µg/mL).

### Plant growth conditions
*Arabidopsis* seedlings were sown in 0.5 X Linsmaier and Skoog nutrient medium (LS) (Caisson Laboratories) and 0.8% w/v agar, stratified at 4 °C for 2 days, and grown in constant light at 22 °C. Phyto agar (PlantMedia/bioWORLD) was used when imaging seedlings and Bacto agar (ThermoFisher) was otherwise.

### Construction and selection of transgenic *Arabidopsis* lines
*Agrobacterium tumefaciens* strain GV3101 was transformed by electroporation, and subsequently grown in LB media with rifampin (Millipore Sigma, 50 µg/mL), gentamicin (Millipore Sigma, 50 µg/mL), any antibiotics carried on the specific plasmid(s), most often kanamycin (Millipore Sigma, 50 µg/mL). The floral dip method[72] was used to generate integrase target lines in Col-0, and then used to introduce each integrase construct into these established target lines. For T1 selection: 120 mg of T1 seeds (-2000 seeds) were sterilized using 70% ethanol and 0.05% Triton-X-100 and then washed using 95% ethanol. Seeds were resuspended in 0.1% agarose and spread onto 0.5X LS Bacto selection plates, using 25 µg/mL of kanamycin for target lines and 25 µg/mL kanamycin and 25 µg/mL hygromycin for lines with both the integrase and the target. The plates were stratified at 4 °C for 48 h then light pulsed for 6 h and covered for 48 h[73]. They were then grown for 4–5 days. To select transformants, tall seedlings with long roots and a vibrant green color were picked from the selection plate with sterilized tweezers and transferred to a new 0.5X LS Phyto agar plate for characterization.

### Characterization of integrase switch in *Arabidopsis* transgenic lines
T1 seedlings for each line were grown 4–5 days after transformant selection. Each selected seedling was imaged at 10X magnification using an epifluorescence microscope (Leica Biosystems, model: DMI 3000) using the RFP (exposure 500 ms, gain 1.6) and CFP (exposure 300 ms, gain 1.6) channels. Selected T1 seedlings were then transferred to soil, and at maturation T2 seeds were selected. For later generations, seedlings were sterilized similarly to T1s, stratified, plated on an LS agar plate, grown for 4–5 days, and characterized using the epifluorescence microscope as for T1.

For the target lines, the seedlings with the highest level of mTurq expression were selected and transferred to soil to generate T2 seeds. The brightest among these lines was maintained as the target line for each integrase, and used for all later transformations of integrase constructs.

For the constitutive integrase constructs in a target line, around 20-30 T1 seedlings were analyzed per construct. Each seedling was categorized into one of five classes as seen in Supplementary Fig. 5

based on the level of switching. Representative images for each category were taken using the RFP and CFP channels and merged for final images. For each construct, the percentage of seedlings in each category was plotted in a bar plot with the number of seedlings tested mentioned at the top of the bar. To evaluate statistical significance, each switching category was assigned a number from 1 through 5 (1 = no switch, 5 = full switch). Significance was determined using analysis of variance (ANOVA) with a post-hoc Tukey's Honest Significance Difference test.

For the YFP to Luciferase PhiC31 target, the target line and the target with pPP2AA3-PhiC31 construct were characterized. T2 seedlings from target lines with and without integrase were grown on LS plates, 7 days old seedlings were imaged with Azure c600 Gel imaging system for YFP fluorescence. Then, 100 μM of Luciferin were sprayed on seedling, after one hour kept in the dark, and seedlings were imaged using NightOwl LB 983 in vivo imager with an exposure time of 10 min.

For the developmental promoter integrase constructs in PhiC31 target line, at least 20 T1 seedlings were analyzed per construct. Each seedling was categorized into one of three classes based on specificity of switching (LR-only, non-exclusive to LR, no-switch). Representative images for each construct were taken using the RFP and CFP channels and merged for final images. A selected number of T1 seedlings with LR-only switch were transplanted to the soil to characterize the T2 generation. For each T1 line, 20 T2 seedlings were characterized in an identical way than for T1s, and similarly for T3 and T4 generations. For each construct, the percentage of seedlings in each of the three categories were plotted in a bar plot with the number of seedlings tested mentioned at the top of the bar.

Python data analysis script which includes statistical tests and plotting functions was run in version 3.9.1 and with the following package dependencies: pandas (version 1.5.3), scipy.stats (version 1.10.0), matplotlib.pyplot (version 3.6.3), matplotlib.colors (version 3.6.3), scikit_posthocs (version 0.21), and numpy (version 1.24.2). All images taken during seedling characterization were opened and processed using the ImageJ program (version 1.53c). Each.tif image file contained the images of a seedling's RFP and CFP channels.tif files were processed through an ImageJ macro to adjust the color lookup table, brightness, and contrast of each channel (RFP: Red, Min: 200, Max: 3000) (CFP: Blue, Min: 200, Max: 4000). After adjustment, the macro overlaid the two channels to create a composite image, rotated the image, added a scale bar, and flattened the image to produce our final processed images.

### Testing the hygromycin resistance of seedlings post characterization

To select T2 hygromycin-resistant seedlings after characterization without selection, the roots of 7 days old seedlings were removed with a razor blade, and seedlings were then transferred onto 0.5X LS BactoAgar plates containing hygromycin. Seedlings were screened for root regrowth after seven days. In our extensive testing of control plants, not all hygromycin resistant seedlings are able to regrow roots after this stressful intervention, but all seedlings that grow roots are truly resistant.

### Characterization of the tuning constructs in *Nicotiana benthamiana*

Integrase target integrated *Nicotiana benthamiana* seeds were acquired from the Orzaez lab[23]. This line has a stably integrated integrase target which switches from LUC firefly luciferase to YFP upon integrase expression. The plants were grown 25 days before injection. *Agrobacterium*-mediated transient transformation of *N. benthamiana* was performed using the *A. tumefaciens* strains GV3101[74]. For each injection in addition to the *A. tumefaciens* with the integrase constructs, we injected an RFP injection efficiency control consisting of constitutively expressed mCherry (donated by Jennifer Brophy) and a construct

containing a P19 gene silencing suppressor protein for enhanced transient transformation[75]. Each *A. tumefaciens* strain was grown overnight in LB at 30 °C, pelleted and incubated in MMA media (10 mM MgCl2, 10 mM MES pH 5.6, 100 μM acetosyringone) for 3 h at room temperature with rotation. Strain density was normalized to an OD600 of 1.5 for each strain in the final mixture of strains before injection. For each integrase construct, the integrase strain, the RFP control, and P19 were injected together; we also injected as control the RFP control and P19 together, as well as the negative control P19 alone. Each *A. tumefaciens* solution was injected into 3–4 different leaves from separate plants. Four days later, hole punches were taken from each injected leaf at 3 locations, and the punches were placed in a 96 well plate. Plate reader measurements of YFP (excitation wavelength: 506 nm, emission wavelength: 541) and mCherry (excitation wavelength: 584 nm, emission wavelength: 610) fluorescence were taken using a Spark® Multimode Microplate Reader by Tecan. Twelve measurements were taken at different locations within the punch. Three tobacco injection replicates per construct were performed and, in each replicate, three leaves were injected. For each punch, the median of the ratio of YFP over RFP fluorescence was calculated and plotted. The box corresponds to the quartile and the median between the different punches for one construct. The data were tested for significance using an ANOVA and post-hoc Tukey's HSD test. The tobacco injection data was plotted and statistically analyzed using a Python data analysis script.

### Confocal imaging of reporter and integrase lines

*Arabidopsis* transgenic reporter lines for *LBD16, ARF19*, and *GATA23* with mScarlet nuclear localized were generated as for integrase switch transgenic lines. After characterization of T1 seedlings, seedlings expressing mScarlet were fixed in 4% formaldehyde using vacuum infiltration followed by ClearSee solution[76]. Fixed and cleared seedlings were mounted on microscope slides using 50% glycerol and Parafilm edges to prevent the coverslips from pressing on the root.

For the integrase lines, for each promoter *LBD16, ARF19*, or *GATA23*, one construct showing a reliable LR-only integrase switch was selected. For each construct, two T1 lines representative of other characterized T1 lines were selected to perform the root bend essay. For each line, 20 T2 seeds of the corresponding T1 line were placed on plates following a specific pattern to avoid a seedling collision after the rotation of the plate. The seeds were stratified for 120 h, grown vertically for 96 h at 22 °C, rotated 90° while keeping the plate vertical, and grown for an additional 20 h. Seedlings were fixed and mounted as mentioned in the previous paragraph.

Imaging of the seedlings were performed using Nikon A1R HD25 laser scanning confocal microscope with 561 laser and 578-623 detector for RFP imaging. For the integrase lines, seedlings were imaged at the bend region, while for the reporter lines, seedlings were scanned to find early-developed lateral roots. Imaging was processed using FIJI. For each imaging, a Z-stack was recorded. First, a maximum average of the Z-stack in the RFP channel was generated. Additionally, we selected one Z-location focusing on the LR nucleus and generated both an image of the RFP channel and the RFP and brightfield merged. The main figure uses the merged RFP/brightfield images.

### Estradiol induction time course

For estradiol induction in T1s, antibiotic selection was performed as described in the method section about *A. thaliana* transgenic lines. Four days after transplanting resistant seedlings onto 0.5X LS Phyto plates, the seedlings are imaged via microscopy in RFP and CFP channels with identical settings as described in the method section about integrase switch seedling characterization. Then the seedlings were transferred onto new 0.5X LS Phyto plates with 10 μM β-estradiol. Each seedling was imaged 24, 48, and 72 h after transplanting onto estradiol and categorized into the appropriate switching category for each timepoint. Data were processed for tuning seedlings.

For estradiol induction in T2s, seeds were plated onto 0.5X LS Phyto plates, stratified for 48 h, and left to grow for 6 days. Then they were transplanted onto 10 μM estradiol 0.5X LS Phyto plates and imaged and categorized for T1 seedlings.

## Reporting summary

Further information on research design is available in the Nature Portfolio Reporting Summary linked to this article.

## Data availability

Data supporting the findings of this work are available within the paper and its Supplementary Information files. A reporting summary for this Article is available as a Supplementary Information file. Plasmids and plant materials are available upon request from JLN (jn7@uw.edu; please expect a response within 3 weeks). Source data are provided with this paper.

## Code availability

Python and ImageJ scripts are available on GitHub [https://doi.org/10.5281/zenodo.7612666].

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

## Acknowledgements

We thank Wesley George, Eric Yang, Dr. Alexander Leydon, as well as other members of the Nemhauser, Imaizumi, and Steinbrenner groups, for feedback and discussions. We thank Eric Yang for developing and gifting the pPP2AA3 promoter; Dr. Jennifer Brophy for sharing the RFP injection efficiency control; members of the Bonnet lab for sending us Bxb1 and Tp901 integrase plasmids; and members of the Orzáez lab for useful discussions, as well as the *N. benthamiana* PhiC31 target line and the YFP to Luc PhiC31 target plasmid. This work was supported by grants from the National Institutes of Health (grant no. GM107084), the National Science Foundation (grant no. IOS-1546873), and the Howard Hughes Medical Institute Faculty Scholars Program. In addition, support to S.G. was provided by EMBO (grant no. ALTF 409-2019).

## Author contributions

S.G. and J.L.N. designed the project. S.G. designed the constructs. C.M. performed and analyzed the tuning and estradiol inducible experiments, S.G. and J.C. performed and analyzed the developmental promoter experiments. S.G., C.M., and J.L.N. wrote the manuscript.

## Competing interests

The authors declare no competing interests.
