## [Peer Review File · Nature Communications]

An integrase toolbox to record gene-expression during plant developmentReviewers' Comments:

Reviewer #1:

Remarks to the Author:

In Guiziou et al, "An integrase toolbox to record gene-expression during plant development", the authors address an exciting problem in the field of synthetic plant biology. The establishment of a visual reporter system to read out the activity of cell type specific promoters over developmental time would be highly valuable to the community of developmental plant biologists. Despite the major advancements in the understanding of cell type specific gene expression over the past decade, the field would benefit from more efficient and versatile systems for lineage tracing and gene expression history. While the system presented by the authors has the potential to be useful to many researchers and represents an honest characterization of a new system, some limitations exist in its current form apparent in the results. These are not discussed enough in the manuscript and appear to be omitted entirely from the abstract. In particular, I think the abstract and the various sections need some discussion of the limitations found in this system and how they compare with existing systems, e.g., CRE-Lox.

These concerns are elaborated below.

1. It is not clear that any of the various iterations of the integrase system presented by the authors functions as needed for lineage tracing in generations beyond T1, with non-target cell types showing integrase activity. Even the destabilized versions of the integrase presented in Figure 4 show considerable variability in specificity between the T1 and T2 generation for all but one promoter used as a test case. Combining this system with a cell type specific inducible promoter may ameliorate this issue, but would require significant testing. Could the authors elaborate in the manuscript on whether the system is ready for use or needs further development?

2. The authors do establish additional utility for this system by generating a split variation of the integrase. However, even when driven with strong, ubiquitous promoters the efficiency of the split system is quite low. It remains an open question whether this will work with cell type specific promoters, which are generally weaker than ubiquitous promoters. Some of the potential obstacles to making the induction system work, which would seem critical to overcoming the issues encountered with markers that invariably express in multiple tissues during development, should be discussed.

3. The authors don't make a strong case for their integrase system relative to existing published systems. There have been a number of publications in plants featuring estradiol-induced integrase systems for lineage/gene history analysis. These come to mind but there may be more:

(Efroni et al. 2016) doi:10.1016/j.cell.2016.04.046

(Smetana et al. 2019) doi:10.1038/s41586-018-0837-0

(Wachsman et al. 2011) doi:10.1105/tpc.111.086199

These aren't cited. The authors do cite a system for synthetic memory circuits in Arabidopsis, which was recently published:

(Lloyd et al. 2022) doi:10.1038/s41587-022-01383-2

There are likely advantages to the integrase system presented here, but the authors should make a clearer argument for the novelty and utility of what they have produced relative to what is already available.

Overall, this has the potential to be a valuable toolset for the plant community. The authors, to their credit, are open about its limitations as currently implemented, and they clearly were rigorous in their testing. These tools are valuable and should be published, but there needs to be a bit more assessment about the performance and readiness of this tool, as well as some discussion of where this

system fits in relation to other tools that have been used in plants.

Reviewer #2:

Remarks to the Author:

The paper of Guiziou et al. describes the development of a technology to plants using orthogonal serine integrases to mediate site-specific and irreversible DNA recombination visualized by switching between fluorescent reporters. The authors do a comprehensive job of describing the development of this integrase toolbox and characterizing its responsiveness both in vitro and in vivo. The work is well presented and I have no further suggestions for this tool being a new and highly useful tool for plant biologists.

Reviewer #3:

Remarks to the Author:

This study by Guiziou et al. reports the development and optimization of integrases for the control of transgene expression in plants. The authors use PhiC31 and Bxb1 integrases to construct gene circuits in Arabidopsis plants, driving integrase expression under the promoters of tissue/cell specific TFs involved in lateral root development and demonstrating conditional integrase activity through reporter expression. A variety of approaches are further tested and applied to tune integrase activity under different promoters, and to improve the control of integrase activity, including degrons, inducible promoters, and split-inteins. The authors also test the stability of the system through generations and in different species. Overall, this is a well performed study, developing valuable tools for plant synthetic biology, to track and potentially manipulate development. The manuscript describes a significant amount of work, despite working in stable transgenic plants, which takes considerable time and effort. In particular, the large sample size and tracking of switch function across generations of plants is particularly impressive. The testing of genetic parts to alter the activity of these integrases is not only useful for integrase circuit development but should be easily applicable to many synthetic biology problems. The transparency of the authors in reporting systems/conditions that show a wide range of efficacy is appreciated, and is very valuable as the field continues to work towards more predictable and robust activity of these types of synthetic systems. It's also really good to see that the code related to this work is available on GitHub and that the plasmid sequences are available in the Supp materials. Overall, this is a really valuable study that extends the toolbox and application of recombinases for control of gene expression in plants. Specific comments on aspects of the study that could be improved are detailed below.

Comments:

- Introduction: "with single-cell methods has led to the discovery of new plant cell types and a more detailed view of cell-fate transitions". This statement should be followed with a citation of Oliva et al. (2022) bioRxiv.

- Introduction: "using integrases, implementing Boolean logic (in bacteria^{15,16}, mammalian cells¹⁷, and plant protoplasts¹⁸". The word "protoplasts" should be removed, to read "plants."

- Introduction: "cell-lineage tracing (in animals¹⁴)" should add "and plants" and cite Efroni et al. (2016).

- Introduction: "We expressed integrases from well-characterized transcription factors essential for lateral root development as a test-case for building synthetic recorders". I think that "transcription factors" should be changed to "transcription factor promoters" or similar meaning.

- Results: "To reduce the likelihood of bidirectional expression from strong promoters like the Cauliflower Mosaic Virus 35S promoter (p35S), and to more closely match the expression level of most developmentally-relevant genes".....could the authors please provide a citation supporting the claim that plant promoters (such as p35S) have bidirectional transcription. Also, why is bidirectional transcription a concern for driving the integrase gene? Fig 1a suggests that it is not proximal to another gene, meanwhile reducing bidirectional transcription from the gene driving mScarlet and mTurquoise2 seems much more important.

- Results: "In all cases, we observed a decrease in LR-only seedlings in the T2 generation (Fig. 3b)." I think this should refer to Fig. 2b.

- Results: "documenting the striking impact of terminator sequences on gene expression⁴²". While the cited paper is a great example, other relevant examples should be included, such as: Damos and Mason (2018) Plant Biotech J, Nagaya et al. (2010) Plant and Cell Physiology, Felippes et al. (2020) The Plant J, and Ingelbrecht et al. (1989) The Plant Cell.

- Results: "we also tested the collection of tuning options with pARF19, another weak promoter known to be expressed in *N. benthamiana* leaves". It is a little strange that the ARF19 promoter was used in *Arabidopsis* as a lateral root specific promoter but in tobacco for leaf transient assays - could the authors please comment on this within the manuscript to make this clear to the reader (and would demonstrate previous differences between *Arabidopsis* and tobacco, supporting other findings within this study).

- Results: "As for the constitutive promoters, we tracked the stability of integrase switching behavior between generations". This could be rephrased as it is awkwardly worded.

- Results: "To analyze later generations, we followed a T1 carrying pGATA23::PhiC31 over four generations, propagating two LR-only seedlings at each generation". It is not clear what tags were used for the examined lines, or if this was a tag free line - please make it explicit what construct was being studied here.

- Results: Fig 4a and b: I think that making the visual separation between the different promoter lines would aid in quick interpretation of this figure, especially for the T2 lines in b, where there are an uneven number of tags examined for each promoter. A box around the 3 designs per promoter in a, and 9/6/3 designs per promoter in b would aid the reader.

- Results: Fig 4c: Please make it explicit which construct design (which tags or no tags) are being examined here in the legend. I assume that it is tag free by comparing T1 and T2 data in the other figure panels but I think making it explicit would be good.

- Results: Fig 5b: please state what is the mScarlet part of the construct as you have in panel a, as you only show what promoter is driving the integrase.

- Methods: The authors should consider submitting plasmid sequences to an online repository, as well as in the supp materials (already done) to allow for ease of access for many users. The Zenodo repository or similar would be appropriate for this and the DOI can be cited in the paper.

- Methods: Are the authors planning to submit plasmids to Addgene? I am sure that there would be interest from other labs in the tools that they have developed and this would aid in open/reproducible science.

Reviewer #4:

Remarks to the Author:

In Guiziou et al, "An integrase toolbox to record gene-expression during plant development", the authors address an exciting problem in the field of synthetic plant biology. The establishment of a visual reporter system to read out the activity of cell type specific promoters over developmental time would be highly valuable to the community of developmental plant biologists. Despite the major advancements in the understanding of cell type specific gene expression over the past decade, the field would benefit from more efficient and versatile systems for lineage tracing and gene expression history. While the system presented by the authors has the potential to be useful to many researchers and represents an honest characterization of a new system, some limitations exist in its current form apparent in the results. These are not discussed enough in the manuscript and appear to be omitted entirely from the abstract. In particular, I think the abstract and the various sections need some discussion of the limitations found in this system and how they compare with existing systems, e.g., CRE-Lox.

These concerns are elaborated below.

1. It is not clear that any of the various iterations of the integrase system presented by the authors functions as needed for lineage tracing in generations beyond T1, with non-target cell types showing integrase activity. Even the destabilized versions of the integrase presented in Figure 4 show considerable variability in specificity between the T1 and T2 generation for all but one promoter used as a test case. Combining this system with a cell type specific inducible promoter may ameliorate this issue, but would require significant testing. Could the authors elaborate in the manuscript on whether the system is ready for use or needs further development?

2. The authors do establish additional utility for this system by generating a split variation of the integrase. However, even when driven with strong, ubiquitous promoters the efficiency of the split system is quite low. It remains an open question whether this will work with cell type specific promoters, which are generally weaker than ubiquitous promoters. Some of the potential obstacles to making the induction system work, which would seem critical to overcoming the issues encountered with markers that invariably express in multiple tissues during development, should be discussed.

3. The authors don't make a strong case for their integrase system relative to existing published systems. There have been a number of publications in plants featuring estradiol-induced integrase systems for lineage/gene history analysis. These come to mind but there may be more:

(Efroni et al. 2016) doi:10.1016/j.cell.2016.04.046
(Smetana et al. 2019) doi:10.1038/s41586-018-0837-0
(Wachsman et al. 2011) doi:10.1105/tpc.111.086199

These aren't cited. The authors do cite a system for synthetic memory circuits in Arabidopsis, which was recently published:

(Lloyd et al. 2022) doi:10.1038/s41587-022-01383-2

There are likely advantages to the integrase system presented here, but the authors should make a clearer argument for the novelty and utility of what they have produced relative to what is already available.

Overall, this has the potential to be a valuable toolset for the plant community. The authors, to their credit, are open about its limitations as currently implemented, and they clearly were rigorous in their testing. These tools are valuable and should be published, but there needs to be a bit more assessment about the performance and readiness of this tool, as well as some discussion of where this system fits in relation to other tools that have been used in plants.

Detailed response to reviews

Reviewer #1 (R1):

While the system presented by the authors has the potential to be useful to many researchers and represents an honest characterization of a new system, some limitations exist in its current form apparent in the results. These are not discussed enough in the manuscript and appear to be omitted entirely from the abstract. In particular, I think the abstract and the various sections need some discussion of the limitations found in this system and how they compare with existing systems, e.g., CRE-Lox.

These concerns are elaborated below.

1. It is not clear that any of the various iterations of the integrase system presented by the authors functions as needed for lineage tracing in generations beyond T1, with non-target cell types showing integrase activity. Even the destabilized versions of the integrase presented in Figure 4 show considerable variability in specificity between the T1 and T2 generation for all but one promoter used as a test case. Combining this system with a cell type specific inducible promoter may ameliorate this issue, but would require significant testing. Could the authors elaborate in the manuscript on whether the system is ready for use or needs further development?

Our response: We thank the reviewer for their positive comments, and for highlighting an area of ambiguity in our description of the integrase toolbox. Our system is ready for use. While we observed variability in the robustness of our integrase switch between independent transgenic lines (a common occurrence in engineering stable lines in plants), we were notably 100% consistent in finding a T1 integrase reporter line that faithfully records the spatiotemporal pattern of the constitutive or developmental promoter being used. For pGATA23 with a SAUR tag, no T2s showed non-specific switching. For pARF19 and pLBD16, we had T1 lines with only 2% or no non-specific switching, respectively.

We find these to be quite robust behaviors for a synthetic device. Indeed, a single robust T1 line per promoter is sufficient for most if not all applications, even given the need to screen in each generation for the appropriate behavior. Most plant synthetic biology papers describe the need for extensive screening to find lines with devices that act according to the design specifications.

For the second point about combining an inducible promoter with a developmental promoter, we fully agree with the reviewer that this may be a necessary step for some developmental promoters of interest. This is why we characterized the XVE system for our toolbox, and showed that it can work quite well. We did not need to use it to

achieve specificity for our developmental promoters, but it is now available for others to deploy as needed for their applications.

We have added language in the discussion highlighting both points.

RI: 2. The authors do establish additional utility for this system by generating a split variation of the integrase. However, even when driven with strong, ubiquitous promoters the efficiency of the split system is quite low. It remains an open question whether this will work with cell type specific promoters, which are generally weaker than ubiquitous promoters. Some of the potential obstacles to making the induction system work, which would seem critical to overcoming the issues encountered with markers that invariably express in multiple tissues during development, should be discussed.

Our response: To address the reviewer's concern about the drop in integrase activity in the split intein system, we have added results from an experiment using the developmental promoter (pARF19) driving both halves of the split intein system (Results section and Supplementary Fig. 10). Splitting the integrase into two parts does lower the integrase activity to some degree, but still works with a developmental promoter. We observed a gain of LR-specific switching in a similar range (a bit better in fact) that what we observed when we combined the SAUR tag with pARF19.

We have added the following language to describe our new results:

“We additionally tested the split-intein integrase system with pARF19 driving both integrase components. We observed LR specific switches in T1 seedlings (FigS10). Even with its lower switching efficiency, the split-intein integrase system allows the recording of developmental gene expression. Using pARF19:Int without tags led to most T2 seedlings having non-specific switching outside of the lateral root. By combining pARF19 with the split-intein system, most of the seedlings now showed the desired LR-specific switching phenotype (Fig. S10b). In this way, the split-intein system can be used as a tuning mechanism, as well as allowing the combination of different promoters.”

RI: 3. The authors don't make a strong case for their integrase system relative to existing published systems. There have been a number of publications in plants featuring estradiol-induced integrase systems for lineage/gene history analysis. These come to mind but there may be more:

(Efroni et al. 2016) doi:10.1016/j.cell.2016.04.046

(Smetana et al. 2019) doi:10.1038/s41586-018-0837-0

(Wachsman et al. 2011) doi:10.1105/tpc.111.086199

These aren't cited. The authors do cite a system for synthetic memory circuits in Arabidopsis, which was recently published:

(Lloyd et al. 2022) doi:10.1038/s41587-022-01383-2

Our response: We thank the reviewer for pointing us to citations that we omitted, and highlighting the need to compare serine and tyrosine integrases—something we had indeed failed to do. To address this oversight, we have added a paragraph in the introduction talking about the tyrosine integrase, underlying studies featuring cell-lineage using tyrosine integrase and differences between serine and tyrosine integrases. We have also added a sentence about estradiol-inducible tyrosine integrase systems when we introduce the characterization of the estradiol-inducible serine integrase.

R1: There are likely advantages to the integrase system presented here, but the authors should make a clearer argument for the novelty and utility of what they have produced relative to what is already available.

Overall, this has the potential to be a valuable toolset for the plant community. The authors, to their credit, are open about its limitations as currently implemented, and they clearly were rigorous in their testing. These tools are valuable and should be published, but there needs to be a bit more assessment about the performance and readiness of this tool, as well as some discussion of where this system fits in relation to other tools that have been used in plants.

Our response: We took the reviewer's critique seriously, and have tried to adjust language in both the introduction and discussion that make it clear what we think are the unique advantages of our toolkit. Namely:

1) Serine integrases can mediate irreversible inversions (tyrosine integrase inversions are reversible), which enables construction of different types of synthetic devices, including order-of-expression transcriptional recorders.

2) Serine integrases work at high efficiency in transient and stable transformation assays, a limitation that has been noted previously for the most common tyrosine integrases (Cre, Flp). They also retain activity over multiple generations, suggesting they are not a frequent target of silencing.

3) We have produced a suite of tools for tuning protein activity that will be of use for diverse applications.

Reviewer #2 (Remarks to the Author) (R2):

The paper of Guiziou et al. describes the development of a technology to plants using orthogonal serine integrases to mediate site-specific and irreversible DNA recombination visualized by switching between fluorescent reporters. The authors do a comprehensive job of describing the development of this integrase toolbox and characterizing its responsiveness both in vitro and in vivo. The work is well presented and I have no further suggestions for this tool being a new and highly useful tool for plant biologists.

Our response: We are grateful for the reviewer's enthusiasm, and share the wish that our work will be of use to many researchers.

Reviewer #3 (Remarks to the Author) (R3):

This study by Guiziou et al. reports the development and optimization of integrases for the control of transgene expression in plants. The authors use PhiC31 and Bxb1 integrases to construct gene circuits in Arabidopsis plants, driving integrase expression under the promoters of tissue/cell specific TFs involved in lateral root development and demonstrating conditional integrase activity through reporter expression. A variety of approaches are further tested and applied to tune integrase activity under different promoters, and to improve the control of integrase activity, including degrons, inducible promoters, and split-inteins. The authors also test the stability of the system through generations and in different species. Overall, this is a well performed study, developing valuable tools for plant synthetic biology, to track and potentially manipulate development. The manuscript describes a significant amount of work, despite working in stable transgenic plants, which takes considerable time and effort. In particular, the large sample size and tracking of switch function across generations of plants is particularly impressive. The testing of genetic parts to alter the activity of these integrases is not only useful for integrase circuit development

but should be easily applicable to many synthetic biology problems. The transparency of the authors in reporting systems/conditions that show a wide range of efficacy is appreciated, and is very valuable as the field continues to work towards more predictable and robust activity of these types of synthetic systems. It's also really good to see that the code related to this work is available on GitHub and that the plasmid sequences are available in the Supp materials. Overall, this is a really valuable study that extends the toolbox and application of recombinases for control of gene expression in plants. Specific comments on aspects of the study that could be improved are detailed below.

Our response: We are grateful for the reviewer's kind comments, as well as for their shared appreciation of the open science practices we have tried to embody in our manuscript.

R3: Comments:

- Introduction: "with single-cell methods has led to the discovery of new plant cell types and a more detailed view of cell-fate transitions". This statement should be followed with a citation of Oliva et al. (2022) bioRxiv.

Our response: We added the citation.

R3:

- Introduction: "using integrases, implementing Boolean logic (in bacteria^{15,16}, mammalian cells¹⁷, and plant protoplasts¹⁸". The word "protoplasts" should be removed, to read "plants."

- Introduction: "cell-lineage tracing (in animals¹⁴)" should add "and plants" and cite Efroni et al. (2016).

Our response: This paragraph in the introduction was referencing the use of serine integrase, we specified this by adding "serine" and added a paragraph mentioning the use of tyrosine integrase in plants and the differences between serine and tyrosine integrases. We chose to leave the word protoplasts, as we think it is useful for readers to know whether a result was from stable or transient transformations.

R3:

- Introduction: "We expressed integrases from well-characterized transcription factors essential for lateral root development as a test-case for building synthetic recorders". I think that "transcription factors" should be changed to "transcription factor promoters" or similar meaning.

Our response: We thank the reviewer for this catch, and modified the text to read "We expressed integrases from promoters of well-characterized transcription factors".

R3:

- Results: "To reduce the likelihood of bidirectional expression from strong promoters like the Cauliflower Mosaic Virus 35S promoter (p35S), and to more closely match the expression level of most developmentally-relevant genes".....could the authors please provide a citation supporting the claim that plant promoters (such as p35S) have bidirectional transcription. Also, why is bidirectional transcription a concern for driving the integrase gene? Fig 1a suggests that it is not proximal to another gene, meanwhile reducing bidirectional transcription from the gene

Our response: Unfortunately, the data on p35S being bidirectional is empirical (for example, it was commonly found as a confounding factor in activation tagging screens where transcripts on either side of an insertion of a T-DNA with p35S enhancer repeats were up-regulated) but we could not find it well-documented in the literature. As

it, thankfully, did not prove to be an issue in our experiments, we removed the first part of the sentence.

R3:

- Results: *“In all cases, we observed a decrease in LR-only seedlings in the T2 generation (Fig. 3b).” I think this should refer to **Fig. 2b**.*

Our response: The reviewer is absolutely correct. We have modified the text accordingly.

R3:

- Results: *“documenting the striking impact of terminator sequences on gene expression⁴²”. While the cited paper is a great example, other relevant examples should be included, such as: Damos and Mason (2018) Plant Biotech J, Nagaya et al. (2010) Plant and Cell Physiology, Felippes et al. (2020) The Plant J, and Ingelbrecht et al. (1989) The Plant Cell.*

Our response: We have added all of the suggested citations.

R3:

- Results: *“we also tested the collection of tuning options with pARF19, another weak promoter known to be expressed in N. benthamiana leaves”. It is a little strange that the ARF19 promoter was used in Arabidopsis as a lateral root specific promoter but in tobacco for leaf transient assays - could the authors please comment on this within the manuscript to make this clear to the reader (and would demonstrate previous differences between Arabidopsis and tobacco, supporting other findings within this study).*

Our response: We have modified the sentence to read: “In addition to the pPP2AA3, p35S, and pUBQ10 promoters, we also tested the collection of tuning options with pARF19, classified here as a weak constitutive promoter. In addition to its well-known role in initiating new lateral roots, ARF19 is also important for leaf expansion in Arabidopsis”.

R3:

- Results: *“As for the constitutive promoters, we tracked the stability of integrase switching behavior between generations”. This could be rephrased as it is awkwardly worded.*

Our response: We modified the sentence to improve clarity: “As for the constructs without tags, we tracked the stability of integrase switching over multiple generations.”

R3:

- Results: "To analyze later generations, we followed a T1 carrying pGATA23::PhiC31 over four generations, propagating two LR-only seedlings at each generation". It is not clear what tags were used for the examined lines, or if this was a tag free line - please make it explicit what construct was being studied here.

Our response: We appreciate the reviewer helping us identify this area of ambiguity in our description of results. Indeed, no tags were added to this line. Throughout the text, we have made this explicit by adding "(no tag)" after pGATA23::PhiC31.

R3:

- Results: Fig 4a and b: I think that making the visual separation between the different promoter lines would aid in quick interpretation of this figure, especially for the T2 lines in b, where there are an uneven number of tags examined for each promoter. A box around the 3 designs per promoter in a, and 9/6/3 designs per promoter in b would aid the reader.

Our response: We thank the reviewer for the suggestion, and have modified the figure accordingly.

R3:

- Results: Fig 4c: Please make it explicit which construct design (which tags or no tags) are being examined here in the legend. I assume that it is tag free by comparing T1 and T2 data in the other figure panels but I think making it explicit would be good.

Our response: As mentioned above, we have now made it clear whenever a construct has no added tags.

R3:

- Results: Fig 5b: please state what is the mScarlet part of the construct as you have in panel a, as you only show what promoter is driving the integrase.

Our response: We added this information to the legend.

R3:

- Methods: The authors should consider submitting plasmid sequences to an online repository, as well as in the supp materials (already done) to allow for ease of access for many users. The Zenodo repository or similar would be appropriate for this and the DOI can be cited in the paper.

Our response: We are depositing all of the plasmids (and their sequences) in AddGene.

R3:

- Methods: Are the authors planning to submit plasmids to Addgene? I am sure that there would be interest from other labs in the tools that they have developed and this would aid in open/reproducible science.

Our response: As mentioned above, we fully agree with the reviewer's perspective. Once our manuscript is accepted and the plasmids are processed by AddGene, they will be accessible with AddGene ID numbers 195885 to 195947. We added a sentence in the methods mentioning it and added the corresponding ID number to the dataset 1 which

lists the constructs.

Reviewers' Comments:

Reviewer #1:

Remarks to the Author:

In this revised version of the manuscript, Guiziou et al. have added additional text that makes the utility of the system more clear and additional data that validate that the split intein system functions with a cell type specific promoter. These additions address the concerns I had on those points.

I remain concerned regarding the stability of the system in generations T2 and beyond. The authors respond that only a single robust T1 line is sufficient for most applications even without functionality in T2, which is what they recovered with two of the three promoters they tested. I am surprised that they argue this is the case, given that T1 lines are grown on selective media, which can dramatically impact plant growth and development. I don't believe this problem should prevent publication, but it would be good if the authors would address this limitation in the text or describe use cases for developmental experiments performed with seedlings grown on selective media. It needs to be clear to a reader how this system can and cannot be used.

Reviewer #3:

Remarks to the Author:

In the revised manuscript the authors have suitably addressed all comments and queries. I look forward to seeing this valuable study in print.

Reviewer #4:

Remarks to the Author:

In this revised version of the manuscript, Guiziou et al. have added additional text that makes the utility of the system more clear and additional data that validate that the split intein system functions with a cell type specific promoter. These additions address the concerns I had on those points.

I remain concerned regarding the stability of the system in generations T2 and beyond. The authors respond that only a single robust T1 line is sufficient for most applications even without functionality in T2, which is what they recovered with two of the three promoters they tested. I am surprised that they argue this is the case, given that T1 lines are grown on selective media, which can dramatically impact plant growth and development. I don't believe this problem should prevent publication, but it would be good if the authors would address this limitation in the text or describe use cases for developmental experiments performed with seedlings grown on selective media. It needs to be clear to a reader how this system can and cannot be used.